# Effects of Using the Surgical Mask and FFP2 during the 6-Min Walking Test. A Randomized Controlled Trial

**DOI:** 10.3390/ijerph182312420

**Published:** 2021-11-25

**Authors:** Sara Cabanillas-Barea, Jacobo Rodríguez-Sanz, Andoni Carrasco-Uribarren, Carlos López-de-Celis, Vanessa González-Rueda, Daniel Zegarra-Chávez, Simón Cedeño-Bermúdez, Albert Pérez-Bellmunt

**Affiliations:** 1Faculty of Medicine and Health Sciences, Universitat International de Catalunya, 08028 Barcelona, Spain; scabanillas@uic.es (S.C.-B.); jrodriguezs@uic.es (J.R.-S.); carlesldc@uic.es (C.L.-d.-C.); vgonzalez@uic.es (V.G.-R.); dzegarra@uic.es (D.Z.-C.); simoncedeno@uic.es (S.C.-B.); aperez@uic.es (A.P.-B.); 2ACTIUM Functional Anatomy Group, 08028 Barcelona, Spain; 3Fundació Institut, Universitari per a La Recerca a l’Atenció, Primària de Salut Jordi Gol i Gurina (IDIAPJGol), 08028 Barcelona, Spain

**Keywords:** walking, dyspnea, N95 respirators, mask, muscle tonus

## Abstract

During the COVID-19 pandemic, the use of masks has been recommended as a containment measure. The mask is a hindrance to normal breathing that causes discomfort. This could put more work on the respiratory accessory muscles, and, consequently, these muscles could see their tone increase. For this reason, during this clinical trial (registered in clincaltrials.gov, number: NCT04789603), it was observed whether the use of the mask produced changes in the distance traveled, in the heart rate, in the oxygenometry, in the self-perceived dyspnea and in the tone of accessory respiratory muscles during a 6-min walk test (6MWT). Fifty healthy volunteers were recruited and carried out the 6MWT on three occasions. They carried out the 6MWT in various situations: using an FFP2/N95 mask, using a surgical mask, and without using a mask. The distance walked, the heart rate, the oxygen therapy, the tone of the accessory respiratory muscles, and the self-perceived dyspnea were recorded in each situation. Significant differences were found between the three situations in terms of self-perceived dyspnea FFP2/N95 > surgical mask > no mask. However, there are no differences between the experimental situations during the 6MWT in terms of distance travelled, heart rate, oxygenometry or respiratory muscle tone.

## 1. Introduction

Since the beginning of the pandemic caused by SARC-CoV 2, more than 259 million cases have been diagnosed and around 5.2 million deaths have been recorded [1]. Currently, a drug is being sought for the treatment of coronavirus [2]. The worldwide effort to create an effective and safe COVID-19 vaccine is beginning to yield results. Several vaccines now have been authorized around the globe; many more remain in development. Nonetheless, in the absence of effective pharmacological treatment and given the virus’s transmission capacity, different alternatives have been proposed to stop the transmission of the virus [2,3,4]. Therefore, these preventive measures against transmission are expected to remain in place for some time.

The transmission of the virus occurs from person-to-person. Different studies conclude that transmission occurs by aerosols from respiratory droplets [2,3,4]. The optimal distance between people to stop person-to-person transmission is uncertain [5]. For this reason and given that there is no effective drug, transmission prevention is of great importance, especially for pandemic mitigation in community settings [6].

The World Health Organization (WHO) established simple precautions to prevent the spread of the virus, such as physical distancing, wearing a mask, keeping rooms well ventilated, avoiding crowds, cleaning your hands, and coughing into a bent elbow or tissue [7]. For this reason, the use of the mask has been proposed in several countries, being mandatory in many of them, for use in the health environment and daily use.

The WHO recommends different types of masks depending on the person, where it will be used, or the population incidence in the area. Wearing a medical/surgical mask is recommended for people over 60, those who have underlying medical conditions, feel unwell, and/or look after an ill family member. For health workers, respirator masks (such as FFP2, FFP3, N95, N99) should be used in settings where procedures are generating aerosols [7]. 

The use of a mask can imply an inspiratory and expiratory restraint and generate a feeling of discomfort in many people. It is responsible for an increase in the inspiratory and expiratory pressures generated [8,9]. This feeling of discomfort and the increment of pressures causes shallow and forced breathing and increases the respiratory accessory musculature activation. 

Person et al. [9] observed that the subjects who used a mask felt dyspnea clinically and significantly higher than those who did not use it. However, to our knowledge, no study has analyzed the parameters of physical effort, respiratory parameters, self-perceived dyspnea, and muscle activation using different types of masks in healthy subjects. For this reason, the present study hypothesis is that there are no changes between wearing and not-wearing a mask (surgical or N-95) in the effort and ventilatory parameters, even though there may be an increase in the tone of the cervical muscles or the perception of dyspnea using a face mask.

This study aims to observe the effect that the surgical mask and the N-95 mask have on the distance walked, the oxygenometry, the heart rate, the sensation of dyspnea, and the tone of the inspiratory accessory muscles during the 6 min walking test (6MWT).

## 2. Materials and Methods

### 2.1. Study Design

A clinical trial was carried out in the Universitat Internacional de Catalunya. A sample of fifty healthy volunteers was recruited from Universitat Internacional de Catalunya. This study was conducted following all ethical protocols (CBAS-2020-13) and Helsinki’s Declaration [10] and was registered at clincaltrials.gov (number: NCT04789603).

A three-group study was conducted (without a mask, with a surgical mask, and with an N95 mask). Each subject was able to participate in all groups. An attempt was made to ensure randomization using computer software (www.random.org (accessed on 20 March 2021)). A researcher, after looking at the number assigned to each volunteer, told each subject which group they belonged to. The CONSORT statement guidelines were used to prepare the study.

### 2.2. Subjects

The study was carried out in the Faculty of Medicine and Health Science, and the sample was recruited during March and May of 2021.

The following points were listed as inclusion criteria: Age more than 18 years, being able to perform the 6MWT, not having cardiac-respiratory diseases, and not having red flags to carry out the study. All participants must have signed the informed consent.

### 2.3. Measurements

The principal outcome of the study was the 6MWT. As secondary outcomes dyspnea using the visual analogue scale (VAS), heart rate (HR) (b/min), transcutaneous oxygen saturation (SpO2) [9,11], and the tone of the accessory respiratory muscles (MyotonPRO) were assessed.

The 6MWT was carried out according to the recommendations of the American Thoracic Society [12]. The volunteers had to walk as fast as possible without running for 6 min. Each minute, the subjects were encouraged to continue at the same pace and not stop. At the end of the test, the heart rate, arterial oxygen saturation, patient self-perceived dyspnea (VAS), and respiratory accessory muscle tone of the patient were measured.

The patients were instructed to place a marker in a horizontal line of 100 mm to indicate how well they could breathe during the 6MWT (i.e., “0” “I could breathe normally” and “100” “I couldn’t breathe”).

The Myoton PRO device was used to register the respiratory accessory muscle tone. This handheld device was designed to assess the muscle and its characteristics. A brief mechanical impulse on the skin generates a natural oscillation of the tissues that cushion it. This tool extracts the numerical values of muscle tension (F- natural oscillation frequency (Hz), characteristic of tone or tension), biomechanical properties (S- dynamic stiffness (N/m) and D- logarithmic decrement of natural oscillation, characteristic of elasticity), and viscoelastic properties (R- mechanical stress relaxation time (ms) and C- ratio of deformation and relaxation time, characteristic of creep) [13,14]. The observed inter-observer ICC of 0.74 and intra-observer ICC of 0.89 [14]. The muscles assessed were the middle scalene and the sternocleidomastoid (SCM).

### 2.4. Procedure

After verifying that the subjects met the inclusion criteria and signed the consent form, they were given a registration number. An investigator (A) observed the number in a random list and included the participant in one of the three groups (without a mask, with a surgical mask, and with an N95 mask). This researcher made an initial registration of demographic data (i.e., gender, age, weight, height, cardiorespiratory pathology, smoker, number of cigarettes per day, a sport performed, hours of daily sport, and days of sport per week). Before the test, all the subjects had to remain for 30 min without a mask, breathing normally. This phase was called the resting phase. Subsequently, the subjects went to the area where the 6MWT was carried out. Each one of them performed the test according to the group to which they had been assigned. The investigator (A) encouraged the participants to take the 6MWT according to the recommendations mentioned above. After performing the 6MWT, all subjects went to the assessment area. All of the subjects wore an N95 mask so that the researcher (B) could not know to which group they had been assigned and had no access to the registration number or the 6MWT test area. This researcher (B) assessed the SpO2 and HR before and after the 6MWT, and after the walking test, assessed the self-perceived dyspnea and the middle scalene and SCM muscle tone. Subjects had the option of retaking the test, repeating all of the phases mentioned above. All participants could participate in all groups if they wanted to.

### 2.5. Statistical Analysis

The statistical analysis was performed with the SPSS Statistics program version 26.0. for Mac. Normal distribution was calculated with the Shapiro–Wilk test (*p* > 0.05). The mean and the standard deviation was calculated. 

For the comparative analysis, the Two-Way Repeated Measures ANOVA test was used. In the case of finding an interaction, the 2 × 3 analysis was performed with Bonferroni correction.

A value of *p* < 0.05 was considered to be statistically significant.

## 3. Results

Fifty healthy volunteers (26 men and 24 women) completed the sample, with a mean age of 20.96 ± 5.36 years. The mean weight was 63.41 ± 10.23 kg and the mean height was 172.38 ± 7.56 cm.

Eighteen percent of the sample were smokers and consumed an average of 3.44 ± 2.00 cigarettes per day. None of the participants reported any cardiorespiratory disease that would prevent them from performing the proposed test. The subjects included in the study performed 3.08 ± 3.57 h a week of physical activity for an average of 2.92 ± 2.19 days a week (Table 1).

No significant difference was found in the distance travelled, in the HR, in the oxygenometry, and the respiratory accessory muscle tone between the different situations analyzed in the study (Table 2). Statistically significant differences were found in the perception of dyspnea (*p* < 0.001) according to the type of mask used. The subjects perceived a higher perception of dyspnea with the FFP2 mask, followed by the surgical mask, and without a mask (Table 3).

## 4. Discussion

The objective of this study was to analyze the differences that occur between not wearing a mask and using a surgical mask or FFP2 in the distance walked in the 6MWT, in HR, in SpO2, in muscular tension of the inspiratory accessory muscles, and in dyspnea perceived due to the use of these masks after performing the 6MWT test.

A sample of fifty healthy volunteers was recruited. This sample resembled, in terms of demographic data, those of the study carried out by Person et al. [9]. The objectives of both studies were very similar. However, the study by Person et al. [9] did not register the muscle tension through Myoton Pro.

### 4.1. Six Minutes of Walking Test

The distance travelled was similar when performing the 6MWT, it was similar for all tests and situations. Depending on the underlying pathology, a change in the distance travelled of 43–54 m during 6MWT is considered to be a clinically significant change [15,16]. The distance covered by the participants was 708.25 ± 77.32 without a mask, 707.50 ± 75.83 with a surgical mask, and 696.09 ± 81.32 with an FFP2 mask. Therefore, the difference between groups never exceeded the minimum difference to consider a significant clinical change. Performing this protocol study in patients with asthma or chronic obstructive pulmonary disease (COPD) may yield different results. When comparing the distance walked during the 6MWT with the study by Person et al. [9], it is observed that both groups of subjects walked practically identical distances. These results cannot be extrapolated to subjects with asthma or other cardiorespiratory diseases such as COPD. Probably, in these subjects, the results would be different.

### 4.2. Heart Rate

The mask is a hindrance for normal expiration and inspiration since there is a barrier in front of the airway. This causes increased respiratory resistance that can increase and prolong inspiratory activity, leading to more negative intrathoracic pressure for longer durations [17]. This prolonged increase in intrathoracic pressure increases cardiac preload and can lead to a higher systolic volume. In addition, the afterload also increases, and everything leads to the myocardium having to consume more oxygen [18,19,20].

The study by Fikenzer et al. [17] concluded that the functional cardiac parameters do not differ significantly at baseline, at maximum load, and during recovery in healthy subjects. However, there is a non-significant trend towards higher cardiac work (Joule) than the no-mask test. This data suggests a myocardial compensation for lung limitation in healthy volunteers. In patients with impaired myocardial function, this compensation may not be possible.

In this study, the HR increased during the 6MWT for all tests and situations, with no statistical changes between them. Studies that seek to observe whether wearing a mask during functional or respiratory tests modifies the heart rate [9,17], have observed similar data to those shown in the present study. HR increased in all three groups after the 6MWT test. The increase in the group without a mask was 20 beats, in the FFP2 group, it was 27 beats, and in the WM group, it was 26 beats. The increase in the group without a mask was lower. This event has also been demonstrated in other studies [9]. However, Roberge et al. [21] showed a greater increase in post-exercise heart rate in the group that used a mask. These differences between the heart rate found between the studies may be due to the test duration as Roberge et al. [21] had a one-hour intervention compared to the 6 min in the present study.

### 4.3. Oxygenometry

One of this study’s purposes was to determine if the use of the FFP2 or SM masks produced changes in blood saturation after performing a routine task, such as walking. It is observed that although there is a tendency towards a change in oxygenation, this occurs in the three study groups, all of which trend in the same direction, a reduction in oxygen in the blood.

Studies that have observed the ventilatory volumes through spirometry have concluded that the masks’ resistance when breathing produces changes in the respiratory rate, tidal volume, and oxygen consumption [17,22]. This increased breathing work does not produce blood oxygen changes, at least in healthy patients, as different studies have shown [9,17,23]. 

### 4.4. Perceived Dyspnea

This variable is the only one that showed statistically significant differences between groups. The group with the highest self-perceived dyspnea was the group with the FFP2 mask and the group with the least self-perceived dyspnea was the group that did not use a mask; the self-perceived dyspnea sensation was FFP2 > SM > WM. Although no studies have been found which compare the three situations (FFP2, SM, WM), there is evidence that subjects who use masks and perform functional tests perceive a greater sensation of dyspnea than those who do not use them.

It has been defined that the use of the mask during moderate exercise contributes to an increase in the temperature and humidity of the inhaled air [21]. The increase in airway temperature is responsible for bronchoconstriction and increased pulmonary resistance during hyperventilation. This fact has been verified in animals and subjects with asthma [24,25]. However, this phenomenon has not been demonstrated in healthy subjects. Our results agree with those of other studies in which the physiological parameters practically do not change with different masks [26]. On the other hand, resistance to breathing, tightness, and general discomfort are the elements with the greatest influence on subjectivity design perception [17,27]. The use of the mask is perceived as disturbing and is accompanied by the increased perception of effort. Therefore, it is likely that masks have a negative impact on perception, especially when exercising [28]. 

All of this contributes to the greater perception of self-perceived dyspnea by the subjects that increases with masks that restrict breathing more. The data presented in this study are consistent with other studies [9,17].

### 4.5. Respiratory Accessory Muscle Tone 

The increased resistance produced by the mask to breathe leads to increased breathing and limitation of ventilation. The increase in respiratory resistance produced by the masks (FFP2 and SM) requires more respiratory muscles than not wearing a mask [17]. To objectify the increase in muscle tone, produced using the mask, the Myoton Pro was used. This tool shows the numerical values of muscle tension and its biomechanical and viscoelastic properties [29]. No statistically significant differences were found in the values of muscle tension, viscoelasticity, and the biomechanical properties provided by Myoton Pro. More studies are needed to assess these values using this tool for a clear conclusion.

The use of masks presents a low economic cost in comparison with other public health measures [30] and has a great benefit for the community. For example, the review conducted by Horward et al. [31] concludes that the widespread use of masks reduces the risk of community transmission of the virus. On the other hand, the early implantation of masks is much more effective compared to delaying their use for reducing transmission; if the mandatory implantation of masks is delayed by 100 days, the impact of their use is minimal [32]. In addition, it seems that using masks reduces the rate of daily infections [33,34]. In light of these insights, it is considered that the use of masks produces a great community benefit and practically no individual harm, as shown in this study.

### 4.6. Limitations

Young people are the main transmitters of COVID-19 and this demographic does not follow the indications for mask use. This is an important limitation of this study due to the fact that it only looks at this type of population. Even so, the observations set the stage to evaluate the effects of masks in other populations, such as subjects with pathologies or the elderly population.

In addition, we have not registered any psychological variables, such as catastrophizing or anxiety. These psychological states could increase the perception of dyspnea, especially when wearing face masks, and could have biased our results. 

Another limitation of the study is that the sample size was not calculated, as there were no previous data to do so. The sample may be small, and perhaps some variables could have reached statistical significance with the suitable sample.

## 5. Conclusions

This study shows that using a mask during aerobic activities, such as the 6MWT, despite influencing the perceived dyspnea, does not modify the distance travelled, the heart rate, the oxygen in the blood, or the inspiratory tone accessory muscles.

## Figures and Tables

**Table 1 ijerph-18-12420-t001:** Characteristics of participants.

Clinical Features	Mean ± SD or n (%) (n = 50)
Sex	
Men	26 (52%)
Women	24 (48%)
Age (years)	20.96 ± 5.36
Height (cm)	172.38 ± 7.56
Weight (Kg)	63.41 ± 10.23
Smoker	
Yes	9 (18%)
No	41 (82%)
Number of cigarettes	3.44 ± 2.00

Abbreviations: SD, Standard Deviation; n, number, %, percentage; cm, centimeters; kg, kilograms; BMI, body mass index.

**Table 2 ijerph-18-12420-t002:** Differences between situations in respiratory accessory muscle tone.

Outcome	Moment	SM	FFP2	WM	F-Value *p*-Value	Outcome	SM	FFP2	WM	F-Value *p*-Value
Right Middle Scalene										
F	PRE	13.88 ± 1.77	14.00 ± 1.27	13.86 ± 1.20	F = 0.025 *p* < 0.975	S	204.14 ± 39.27	217.71 ± 50.75	209.80 ± 29.48	F = 2.221 *p* < 0.118
	POST	13.69 ± 1.35	13.79 ± 1.52	13.72 ± 1.17			210.5 ± 44.98	204.98 ± 39.95	210.46 ± 40.96	
D	PRE	1.03 ± 0.17	1.00 ± 0.19	1.04 ± 0.16	F = 1.476 *p* < 0.234	R	21.84 ± 2.95	21.17 ± 2.98	21.50 ± 2.90	F = 2.538 *p* < 0.089
	POST	1.05 ± 0.16	1.05 ± 0.21	1.04 ± 0.17			21.45 ± 3.43	22.08 ± 3.20	21.85 ± 3.14	
C	PRE	1.25 ± 0.22	1.22 ± 0.19	1.23 ± 0.18	F = 0.499 *p* < 0.608					
	POST	1.23 ± 0.17	1.23 ± 0.21	1.25 ± 0.17						
Right SCM										
F	PRE	12.50 ± 0.79	12.72 ± 1.31	12.52 ± 0.67	F = 1.127 *p* < 0.328	S	186.12 ± 22.36	182.31 ± 25.27	186.00 ± 16.19	F = 2.520 *p* < 0.119
	POST	1.23 ± 0.17	1.23 ± 0.21	1.25 ± 0.17			186.52 ± 17.02	182.71 ± 18.39	187.64 ± 15.34	
D	PRE	1.15 ± 0.22	1.11 ± 0.19	1.13 ± 0.22	F = 0.997 *p* < 0.373	R	25.65 ± 2.37	25.32 ± 2.98	25.54 ± 2.23	F = 0.218 *p* < 0.804
	POST	1.16 ± 0.24	1.17 ± 0.21	1.15 ± 0.21			25.94 ± 2.33	25.89 ± 2.98	25.72 ± 2.83	
C	PRE	1.50 ± 0.17	1.50 ± 0.24	1.49 ± 0.18	F = 0.070 *p* < 0.926					
	POST	1.50 ± 0.17	1.49 ± 0.18	1.49 ± 0.18						
Left Middle Scalene										
F	PRE	13.80 ± 0.97	13.72 ± 0.89	13.63 ± 1.07	F = 0.363 *p* < 0.696	S	205.36 ± 30.17	206.96 ± 34.14	208.02 ± 33.37	F = 0.689 *p* < 0.497
	POST	13.67 ± 1.03	13.78 ± 1.11	13.58 ± 0.95			207.48 ± 33.01	213.08 ± 32.17	207.10 ± 28.51	
D	PRE	1.00 ± 0.14	1.01 ± 0.19	1.02 ± 0.15	F = 1.000 *p* < 0.372	R	20.84 ± 2.64	21.12 ± 2.42	20.98 ± 2.70	F = 1.000 *p* < 0.322
	POST	1.03 ± 0.13	1.04 ± 0.17	1.02 ± 0.12			21.13 ± 2.29	21.25 ± 2.79	21.46 ± 2.63	
C	PRE	1.17 ± 0.14	1.18 ± 0.15	1.22 ± 0.25	F = 1.122 *p* < 0.326					
	POST	1.19 ± 0.15	1.20 ± 0.16	1.20 ± 0.16						
Left SCM										
F	PRE	12.71 ± 0.91	12.86 ± 1.03	13.02 ± 1.51	F = 1.000 *p* < 0.372	S	190.00 ± 21.41	188.65 ± 24.87	189.16 ± 23.38	F = 0.208 *p* < 0.797
	POST	12.87 ± 1.22	12.64 ± 0.86	12.83 ± 0.79			186.60 ± 30.19	189.22 ± 26.23	193.58 ± 25.99	
D	PRE	1.13 ± 0.20	1.10 ± 0.25	1.11 ± 0.18	F = 0.184 *p* < 0.832	R	24.82 ± 2.33	24.28 ± 2.19	24.38 ± 2.12	F = 1.000 *p* < 0.322
	POST	1.12 ± 0.17	1.11 ± 0.17	1.12 ± 0.19			24.82 ± 2.40	25.20 ± 2.66	24.23 ± 2.04	
C	PRE	1.42 ± 0.15	1.39 ± 0.16	1.39 ± 0.14	F = 2.600 *p* < 0.080					
	POST	1.43 ± 0.16	1.43 ± 0.20	1.39 ± 0.14						

Abbreviations: SCM, sternocleidomastoid; SM, with surgical mask; FFP2, with FFP2 mask; WM, without mask. F, natural oscillation frequency; S, dynamic stiffness; D, characterize of elasticity; R, mechanical stress relaxation time; C, ratio of deformation and relaxation time.

**Table 3 ijerph-18-12420-t003:** Differences between situations in HR, SpO2, Distance, and Dyspnea sensation.

Outcome	Moment	SM	FFP2	WM	F-Value *p*-Value
HR	PRE	79.52 ± 16.99	77.84 ± 16.28	80.70 ± 15.46	F = 1.302 *p* < 0.277
	POST	99.86 ± 25.74	105.67 ± 32.20	107.24 ± 33.53	
SpO_2_	PRE	97.84 ± 1.49	97.37 ± 2.73	96.62 ± 3.68	F = 2.732 *p* < 0.070
	POST	94.88 ± 8.22	96.94 ± 2.60	95.88 ± 4.74	
Distance (m)		707.50 ± 75.83	696.09 ± 81.32	708.25 ± 77.32	F = 0.202 *p* < 0.831
Dyspnea sensation	POST	2.48 ± 1.71	3.52 ± 1.97	1.54 ± 1.69	F = 3.419 *p* < 0.001

Abbreviations: HR, heart rate; SM, surgical mask; FFP2, FFP2 mask; WM, without mask.

## Data Availability

You can find all the data from this study in the “HARVARD Dataverse” using the following link: https://dataverse.harvard.edu/dataset.xhtml?persistentId=doi:10.7910/DVN/V7X9JH (accessed on 24 April 2021).

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
