# Peer review of "Effects of Using the Surgical Mask and FFP2 during the 6-Min Walking Test. A Randomized Controlled Trial"

_ijerph, 2021, doi:10.3390/ijerph182312420_

Round 1
Reviewer 1 Report
I was able to read the work presented by the authors with great attention and interest.
The paper shows that no statistically significant difference is present between the groups observed in the study, namely WM, FFP2 and SM. In particular, there are no significant differences in the main characteristic outcomes of 6MWT, namely HR, SpO2, Distance and in addition even in the respiratory accessory muscle tone. The only significant difference between the groups evaluated was in the perception of dyspnea using visual analogue scale (VAS).
Two important references are cited in the text of the paper (9 and 17). From these references it is possible to assess that it is already known that the use of SM and FFP2 masks does not modify the main outcomes of the 6MWT and that the use of the different masks only affects the perception of dyspnea. Furthermore, the increase in humidity, as already known, can cause bronchoconstriction and increase in resistance during hyperventilation; this could be a problem during 6MWT and cardiopulmonary exercise test especially in patients with obstructive diseases (Asthma and COPD).
Furthermore, from the results, what is statistically significant is relative to a personal deduction of dyspnea. Was it considered that the patient, using the VAS, was implicitly to respond worse if he wore the mask precisely because of a psychological question of claustrophobic sensations or anxiety?
It would have been interesting to understand how the patients were trained to give the answers. In a nutshell, can a data of perception alone represent a useful result in scientific research, especially for a test that is normally used for multiple obstructive or restrictive pathologies of the respiratory system?
For these reasons, I do not believe that the data obtained can represent or contribute to scientific research and do not determine a scientific and clinical impact (among other things for non-original data but presented in other papers already published).
Furthermore, the text lacks a summary table of the characteristics of the patients that allow to understand well what are the characteristics of the patients and the comorbid events (in the paper it is clear that the patients are smokers and that they do not have particular cardiovascular diseases, but they are state excludes only serious heart diseases ??? Other comorbidities???)
It would have been interesting, stratifying the patients by pathology, not using apparently healthy patients, to understand the effect of using the surgical mask or FFP2 on the 6MWT outcomes (in particular for patients with COPD, bronchiectasis, asthma).
Author Response
- I was able to read the work presented by the authors with great attention and interest.
The paper shows that no statistically significant difference is present between the groups observed in the study, namely WM, FFP2 and SM. In particular, there are no significant differences in the main characteristic outcomes of 6MWT, namely HR, SpO2, Distance and in addition even in the respiratory accessory muscle tone. The only significant difference between the groups evaluated was in the perception of dyspnea using visual analogue scale (VAS).
Two important references are cited in the text of the paper (9 and 17). From these references it is possible to assess that it is already known that the use of SM and FFP2 masks does not modify the main outcomes of the 6MWT and that the use of the different masks only affects the perception of dyspnea. Furthermore, the increase in humidity, as already known, can cause bronchoconstriction and increase in resistance during hyperventilation; this could be a problem during 6MWT and cardiopulmonary exercise test especially in patients with obstructive diseases (Asthma and COPD).
Furthermore, from the results, what is statistically significant is relative to a personal deduction of dyspnea. Was it considered that the patient, using the VAS, was implicitly to respond worse if he wore the mask precisely because of a psychological question of claustrophobic sensations or anxiety?It would have been interesting to understand how the patients were trained to give the answers. In a nutshell, can a data of perception alone represent a useful result in scientific research, especially for a test that is normally used for multiple obstructive or restrictive pathologies of the respiratory system?
For these reasons, I do not believe that the data obtained can represent or contribute to scientific research and do not determine a scientific and clinical impact (among other things for non-original data but presented in other papers already published).
- As the reviewer points out, the use of a mask and the VAS recording of self-perceived dyspnea may be conditioned by psychological issues (line 263-265).Therefore, it could be considered a limitation and we have reflected this in the limitations. In addition, information on how this variable was recorded in the study has been added (line 100-101).
- Furthermore, the text lacks a summary table of the characteristics of the patients that allow to understand well what are the characteristics of the patients and the comorbid events (in the paper it is clear that the patients are smokers and that they do not have particular cardiovascular diseases, but they are state excludes only serious heart diseases ??? Other comorbidities???)
A table with descriptive sample data has been included in the manuscript as suggested by the reviewers.
The sentence “None of the participants had any severe cardiorespiratory disease” line 144 has been changed by “None of the participants reported any cardiorespiratory disease that would prevent them from performing the proposed test.”.
- It would have been interesting, stratifying the patients by pathology, not using apparently healthy patients, to understand the effect of using the surgical mask or FFP2 on the 6MWT outcomes (in particular for patients with COPD, bronchiectasis, asthma).
Our subjects did not suffer from any cardiorespiratory disease. Therefore, we cannot stratify the data. A sentence has been added saying that the data cannot be extrapolated to subjects with cardiopulmonary diseases (line 177-179).
Reviewer 2 Report
The study aims to assess the impact of using different types of masks on the ability to walk and breathe in the general population using the 6MWT as a performance test. The study is scientifically sound, but I have some comments for the authors that need to be addressed to clarify things that were presented in the manuscript:
- In the Methods section, the sample size calculation was not provided and therefore some of the differences that were seen between the groups could have been significant if the authors had a large enough sample for these outcomes.
- In the Results section,
- I would like to see a table presenting the participants' demographics. Also, no units were provided for the height and weight.
- For the smoking variable, the mean number of cigarettes (among smokers) was below one, can you explain why?
- The sentence "The subjects perceived a higher perception of dyspnea with the FPP2 mask, followed by the surgical mask, and without a mask." implies that this order is based on statistical significance between the groups. However, this needs to be supported by post-hoc analyses to identify which differences between the pairs were actually signficant.
- In the Discussion section,
- The paragraph "... the mask does not modify the distance travelled during this test. The distance covered by the participants was 708.25 ± 77.32 without a mask, 707.50±75.83 with a surgical mask, and 696.09 ± 81.32 with an FFP2 mask. Although no statistically significant differences were shown, we observed a slight decrease in the distance travelled when the subjects wore a mask, the difference being greater when with the FFP2 mask. The clinically relevant difference for 6MWT is 43-54m depending on the participants' pathology [15,16]." Since the differences between the distances were very small and much lower than the clinically relevant differences (as this and previous studies have concluded), I recommend that the authors remove the emphasis on this small difference and focus on the impact of this result and how it could be different if a different population was studied.
- In the sentence: "The data showed that after the 6MWT test without a mask, the heart rate decreased less than using a mask, this has also been shown in other studies [9]", the heart rate actually increased in all groups, so I am not sure what the authors are referring to in this sentence.
- In the Conclusion section,
- The authors need to be less assertive about their findings. Since no actual influence on the clinical measures assessed was observed between the groups, I don't agree that any "significant clinical influence" should be indicated in the conclusion. Perception is more of a psychological measure than a clinical measure, therefore the authors should revise the conclusion section.
- Also, repeating the main findings in the conclusion section is not the purpose of this section. The authors should focus on the impact of their results on the public, patients with other conditions in future studies, ... etc.
Minor comment:
- FFP2 was incorrectly typed on multiple occasions in this manuscript.
Author Response
The study aims to assess the impact of using different types of masks on the ability to walk and breathe in the general population using the 6MWT as a performance test. The study is scientifically sound, but I have some comments for the authors that need to be addressed to clarify things that were presented in the manuscript:
- In theMethods section, the sample size calculation was not provided and therefore some of the differences that were seen between the groups could have been significant if the authors had a large enough sample for these outcomes.
We have this sentence “Another limitation of the study is that the sample size was not calculated, as there was no previous data to do so. The sample may be small, and perhaps some variables could have reached statistical significance with the suitable sample.” In the limitation section.
- In the Results section, I would like to see a table presenting the participants' demographics. Also, no units were provided for the height and weight.
As suggested by the reviewers, a table with descriptive sample data and comorbidity data has been included in the manuscript. In addition, units of measurement for weight and height have been included in the text.
- For the smoking variable, the mean number of cigarettes (among smokers) was below one, can you explain why?
The error in the manuscript has been corrected. The data shown corresponded to the mean of the entire sample and not only of the smoking subjects.
- The sentence "The subjects perceived a higher perception of dyspnea with the FPP2 mask, followed by the surgical mask, and without a mask." implies that this order is based on statistical significance between the groups. However, this needs to be supported by post-hoc analyses to identify which differences between the pairs were actually signficant.
This statement is based on the subjects' mean scores on their perception of breathlessness, within each situation being statistically significant with Bonferroni's post-hoc in all comparisons. It also coincides with the mean distance travelled by the subjects, although there is no statistically significant difference in this case. It also coincides with the more excellent protection offered by each situation.
- In the Discussion section,
The paragraph "... the mask does not modify the distance travelled during this test. The distance covered by the participants was 708.25 ± 77.32 without a mask, 707.50±75.83 with a surgical mask, and 696.09 ± 81.32 with an FFP2 mask. Although no statistically significant differences were shown, we observed a slight decrease in the distance travelled when the subjects wore a mask, the difference being greater when with the FFP2 mask. The clinically relevant difference for 6MWT is 43-54m depending on the participants' pathology [15,16]." Since the differences between the distances were very small and much lower than the clinically relevant differences (as this and previous studies have concluded), I recommend that the authors remove the emphasis on this small difference and focus on the impact of this result and how it could be different if a different population was studied.
We changed some information in this paragraph “The distance travelled was similar when performing the 6MWT, it was similar for all tests and situations. Therefore, the mask does not modify the distance travelled during this test. The distance covered by the participants was 708.25 ± 77.32 without a mask, 707.50±75.83 with a surgical mask, and 696.09 ± 81.32 with an FFP2 mask. No statistically or clinically significant differences were found between groups. It may be that performing this protocol study in patients with asthma or chronic obstructive pulmonary disease (COPD) will show different results. The clinically relevant difference for 6MWT is 43-54m depending on the participants' pathology [15,16].”
In the sentence: "The data showed that after the 6MWT test without a mask, the heart rate decreased less than using a mask, this has also been shown in other studies [9]", the heart rate actually increased in all groups, so I am not sure what the authors are referring to in this sentence.
We have added information in the sentence so that it is better understood “The data showed that after the 6MWT test without a mask, the heart rate decreased less than using the surgical or the FFP2 mask, this has also been shown in other studies [9]”
- In the Conclusion section,
The authors need to be less assertive about their findings. Since no actual influence on the clinical measures assessed was observed between the groups, I don't agree that any "significant clinical influence" should be indicated in the conclusion. Perception is more of a psychological measure than a clinical measure, therefore the authors should revise the conclusion section.
We have changed the conclusion by “This study shows that using a mask during aerobic activities such as the 6MWT, despite influencing the perceived dyspnea, does not modify the distance travelled, the heart rate, the oxygen in the blood, or the inspiratory tone accessory muscles.
Also, repeating the main findings in the conclusion section is not the purpose of this section. The authors should focus on the impact of their results on the public, patients with other conditions in future studies, ... etc.
We have changed the conclusion by “This study shows that using a mask during aerobic activities such as the 6MWT, despite influencing the perceived dyspnea, does not modify the distance travelled, the heart rate, the oxygen in the blood, or the inspiratory tone accessory muscles.
Minor comment:
FFP2 was incorrectly typed on multiple occasions in this manuscript.
The term has been unified throughout the text
Reviewer 3 Report
The work discusses the differences in different types of masks (no mask, surgical mask, FFp2) on their effects in various parameters. Given that the work do not receive any external fundings, it is commendable in its own right. However, there are numerous points in the article that warrants attention:
The abstract appears to follow the template of some other journals. I believe a coherent single paragraph outline should be included instead.
Given that the work discusses the effects of mask wearing, I believe additional citations should be included in the introduction, such as the following:
- https://www.mdpi.com/1660-4601/18/17/9027
- https://www.pnas.org/content/117/51/32293
- https://www.sciencedirect.com/science/article/pii/S2590113320300201?via%3Dihub
- https://www.healthaffairs.org/doi/10.1377/hlthaff.2020.00818
- https://www.pnas.org/content/118/4/e2014564118
Line 71: The acronym 6MWT should be included the first time it appears in the main text.
Line 87: replace ">" with "more than"
Line 98: who reports the self-perceived dyspnea, the investigators or the participants
Line 120: Why is that step needed? How did the investigator ensures that he/she is blinded to the subsequent assessment?
Line 132-137: why were Shapiro-Wilk test and Two-Way repeated measures ANOVA chosen? Would appreciate elaboration.
Lines 238-241: I could not see why the limitations mentioned limits the study itself?
Line 244: did you mean 6MWT?
While the study is interesting in it's own right, it appeared to me that the work at this stage lack the necessary discussions for the readers to appreciate the significance of the work.
Author Response
- The work discusses the differences in different types of masks (no mask, surgical mask, FFp2) on their effects in various parameters. Given that the work do not receive any external fundings, it is commendable in its own right. However, there are numerous points in the article that warrants attention:
The abstract appears to follow the template of some other journals. I believe a coherent single paragraph outline should be included instead.
We have changed the abstract paragraph:
“During the COVID-19 pandemic, the use of masks has been recommended as a containment measure. The mask is a hindrance to normal breathing that causes discomfort. This could put more work on the respiratory accessory muscles, and, consequently, these muscles could see their tone increase. For this reason, during this clinical trial (registered in clincaltrials.gov, number: NCT04789603), it was observed whether the use of the mask produced changes in the distance traveled, in the heart rate, in the oxygenometry, in the self-perceived dyspnea and in the Tone of accessory respiratory muscles during the 6-minute walk test (6MWT). Fifty healthy volunteers were recruited and carried out the 6MWT on three occasions. They were exposed to the situations using an FFp2/N95 mask, a surgical mask, and the test without a mask. The distance walked, the heart rate, the oxygen therapy, the tone of the accessory respiratory muscles, and the self-perceived dyspnea were recorded in each situation. Significant differences were found between three situations of self-perceived dyspnea FFp2 / N95> Surgical mask> No mask. However, there are no differences between the experimental situations during the 6MWT in distance travelled, heart rate, oxygenometry or respiratory muscle tone.”
- Given that the work discusses the effects of mask wearing, I believe additional citations should be included in the introduction, such as the following:
- https://www.mdpi.com/1660-4601/18/17/9027
- https://www.pnas.org/content/117/51/32293
- https://www.sciencedirect.com/science/article/pii/S2590113320300201?via%3Dihub
- https://www.healthaffairs.org/doi/10.1377/hlthaff.2020.00818
- https://www.pnas.org/content/118/4/e2014564118
We have reviewed the recommended references for our text, and we have found that they are adequate, so we added this additional reference (reference 29-33)
- Line 71: The acronym 6MWT should be included the first time it appears in the main text
The acronym has been included the first time it appears in the text.
- Line 87: replace ">" with "more than".
We have replace “>” for “more than”
- Line 98: who reports the self-perceived dyspnea, the investigators or the participants
We have clarified this point in the text including “patient self-perceived…”
- Line 120: Why is that step needed? How did the investigator ensures that he/she is blinded to the subsequent assessment?
The paragraph has been changed. An attempt has been made to improve the wording and give importance to how investigator B was blinded. Line 112-130
- Line 132-137: why were Shapiro-Wilk test and Two-Way repeated measures ANOVA chosen? Would appreciate elaboration.
The Shapiro-Wilk test is a specific test for normality, whereas the method used by Kolmogorov-Smirnov test is more general, but less powerful. We would indeed be right at the limit, and we could also have used the Kolmogorov-Smirnov test, but the Shapiro Wilk test was chosen. The Shapiro–Wilk test is a more appropriate method for small sample sizes, although it can also handle larger sample sizes.
A Two-Way Repeated Measures ANOVA was chosen as a robust test. The objective was to observe the interaction between the Time factor and the Face Mask factor.
8.Lines 238-241: I could not see why the limitations mentioned limits the study itself?
We have added new limitations
- Line 244: did you mean 6MWT?
We have corrected this mistake
- While the study is interesting in it's own right, it appeared to me that the work at this stage lack the necessary discussions for the readers to appreciate the significance of the work
We have added a new paragraph at the discussion
“The use of the mask has a low economic cost in comparison with other public health measures [29] and has a great benefit for the community. For example, the re-view conducted by Horward et al. [30] concludes that the widespread use of the mask reduces the risk of community transmission of the virus. On the other hand, the early implantation of the mask is much more effective compared to delaying its use for reducing the transmission, if the mandatory implantation of the mask is delayed 100 days, the impact of its use is minimal [31]. In addition, it seems that using it reduces the rate of daily infections [32,33]. For all this, it is considered that its use produces a great community benefit and practically no individual harm as shown in this study.”
Round 2
Reviewer 1 Report
I reassessed the paper and I believe it has reached a sufficient scientific level.
Author Response
Hemos agregado los cambios sugeridos por otros revisores.
Reviewer 2 Report
Thank you for addressing the comments. I just have a few comments that still need to be revised for improvement:
- In the following sentence: "The distance covered by the participants was 708.25 ± 77.32 without a mask, 168 707.50 ± 75.83 with a surgical mask, and 696.09 ± 81.32 with an FFP2 mask. No statistically or clinically significant differences were found between groups. It may be that performing this protocol study in patients with asthma or chronic obstructive pulmonary disease (COPD) will show different results. The clinically relevant difference for 6MWT is 43-54m depending on the participants' pathology [15,16]." The idea is acceptable now but the order of these sentences needs to be revised to improve the readability of this section.
- For the following sentence: "The data showed that after the 6MWT test without a mask, the heart rate decreased less than using the surgical or the FFP2 mask, this has 198 also been shown in other studies [9]." This sentence still needs to be revised with extra attention.
- For the comment "FFP2 was incorrectly typed on multiple occasions in this manuscript", still, there are many places where FFP2 was typed incorrectly as "FPP2".
Author Response
Thank you for addressing the comments. I just have a few comments that still need to be revised for improvement:
- In the following sentence: "The distance covered by the participants was 708.25 ± 77.32 without a mask, 707.50 ± 75.83 with a surgical mask, and 696.09 ± 81.32 with an FFP2 mask. No statistically or clinically significant differences were found between groups. It may be that performing this protocol study in patients with asthma or chronic obstructive pulmonary disease (COPD) will show different results. The clinically relevant difference for 6MWT is 43-54m depending on the participants' pathology [15,16]." The idea is acceptable now but the order of these sentences needs to be revised to improve the readability of this section.
This sentence has been changed by “Depending on the underlying pathology, a change in distance travelled of 43-54 m during 6MWT is considered a clinically significant change [15,16]. The distance covered by the participants was 708.25 ± 77.32 without a mask, 707.50 ± 75.83 with a surgical mask, and 696.09 ± 81.32 with an FFP2 mask. Therefore, the difference between groups never exceeded the minimum difference to consider a significant clinical change. Performing this protocol study in patients with asthma or chronic obstructive pulmonary disease (COPD) may yield different results.”
2. For the following sentence: "The data showed that after the 6MWT test without a mask, the heart rate decreased less than using the surgical or the FFP2 mask, this has 198 also been shown in other studies [9]." This sentence still needs to be revised with extra attention.
This sentence has been changed by “HR increased in all three groups after the 6MWT test. The increase in the group without a mask was 20 beats, in the FFP2 group, it was 27 beats, and in the WM group, it was 26 beats. The increase in the group without a mask was lower. This event has also been demonstrated in other studies”
3. For the comment "FFP2 was incorrectly typed on multiple occasions in this manuscript", still, there are many places where FFP2 was typed incorrectly as "FPP2".
We have replaced FPP2 for FFP2.

Reviewer 3 Report
The current version is a significant improvement from the first with much greater clarity. My only concern, albeit relatively major, would be the significance of self-perceived dyspnea on the type of masks, especially that it has been well-understood and explored. I noted that the authors have added the lines 222 to 230, but I think it is inadequate to highlight the significance of this work, especially given that the difference in self-perceived dyspnea is central to this work. I would strongly suggest the authors to compare existing works and highlight explicitly how their work adds value to the existing literature.
Author Response
The current version is a significant improvement from the first with much greater clarity. My only concern, albeit relatively major, would be the significance of self-perceived dyspnea on the type of masks, especially that it has been well-understood and explored. I noted that the authors have added the lines 222 to 230, but I think it is inadequate to highlight the significance of this work, especially given that the difference in self-perceived dyspnea is central to this work. I would strongly suggest the authors to compare existing works and highlight explicitly how their work adds value to the existing literature.
Response: New information has been added at this paragraph “However, this phenomenon has not been demonstrated in healthy subjects. Our results agree with those of other studies in which the physiological parameters practically do not change with different masks.” Line 228-230